# Climate Change, Environmental Health, and Challenges for Nursing Discipline

**DOI:** 10.3390/ijerph20095682

**Published:** 2023-04-28

**Authors:** Omar Portela Dos Santos, Pauline Melly, Stéphane Joost, Henk Verloo

**Affiliations:** 1Department of Nursing Sciences, School of Health Sciences, HES-SO Valais/Wallis, University of Applied Sciences and Arts Western Switzerland, CH-1950 Sion, Switzerland; pauline.melly@hevs.ch (P.M.); henk.verloo@hevs.ch (H.V.); 2Institute of Health Sciences, Universidade Católica Portuguesa, 4169-005 Porto, Portugal; 3Geospatial Molecular Epidemiology Group (GEOME), Laboratory for Biological Geochemistry (LGB), Ecole Polytechnique Fédérale de Lausanne (EPFL), CH-1015 Lausanne, Switzerland; stephane.joost@epfl.ch; 4Service of Old Age Psychiatry, Department of Psychiatry, Lausanne University Hospital, CH-1008 Lausanne, Switzerland

**Keywords:** climate change, environmental health, planetary health, nursing discipline, sustainability, metaparadigm, environment, nursing interventions, health promotion, disease prevention

## Abstract

Current data and scientific predictions about the consequences of climate change are accurate in suggesting disaster. Since 2019, climate change has become a threat to human health, and major consequences on health and health systems are already observed. Climate change is a central concern for the nursing discipline, even though nursing theorists’ understanding of the environment has led to problematic gaps that impact the current context. Today, nursing discipline is facing new challenges. Nurses are strategically placed to respond to the impacts of climate change through their practice, research, and training in developing, implementing, and sustaining innovation towards climate change mitigation and adaptation. It is urgent for them to adapt their practice to this reality to become agents of change.

## 1. Introduction

In the 1950s, the nursing discipline evolved from a vocation to an academic discipline—a shift that required the development of its own vision and knowledge [1]. Barbara Carper [2] expressed this knowledge formally by developing the concept of nursing knowledge and its four fundamental patterns of knowing, one of which is esthetic knowing. This is defined as the art of caring, and it is fundamental to creating a care environment that is both therapeutic and caring for patients. The present essay uses a historical approach to explore nursing’s place—the art of care’s place—in the face of climate change. The environment will be addressed through the lens of Florence Nightingale’s environmental theory, Hardy’s and Fawcett’s opposing visions about the metaparadigm, the three paradigms of categorization, integration, and transformation, and an analysis of the concepts of exposure in environmental health and sustainability. Next, climate change and its associated health, social, and political consequences will be outlined. The implication of the nursing discipline will also be exposed through a bibliometric analysis. Finally, the pragmatic role that the nursing discipline must adopt in the face of climate change will be highlighted through practice, research, and implementation interventions according to two theoretical frameworks: the conceptual model of enhancing equity and quality and implementation science.

### 1.1. Today’s Environment: Climate Challenges

In 1995, the Institute of Medicine (IOM) defined environmental health as the absence of illness or injury despite chemical, physical, or biological environmental hazards [3]. Current data and scientific predictions about the consequences of climate change are accurate in suggesting disaster [4,5]. Between 2030 and 2050, climate change is expected to cause about 250,000 additional deaths per year [6]. Since 2019, it has become a recognized global threat to human health, along with non-communicable and cardiovascular diseases [7] Its main effects on population health [8,9,10] are increases in mental disorders, heart failure, chronic kidney disease, asthma, allergies, chronic obstructive pulmonary disease, cancer, myocardial infarction, and stroke. Environmental risk factors account for 80% of diseases [11] and 25–30% of the total disease burden [7,12]. 

Rising average temperatures and sea levels [5], exposure to sulfur dioxide (SO_2_), nitrogen dioxide (NO_2_), various fine suspended particles (PM_2.5_ and PM_10_), and modifications to the tropospheric ozone (O_3_) layer are the main consequences of climate change [8]. Europe is one of the main contributors to the climate crisis, with its healthcare sector alone emitting 4.4% of all global emissions. If it were a country, it would be the fifth largest emitter on the planet [13].

In Europe, since 1850, the average air temperatures have risen by 2.2 °C, about 1 °C more than the estimated global temperature increase of 1.2 °C. Since 2010, the population’s exposure to heat waves has increased by 57%, putting vulnerable people (older adults, children, people with chronic conditions, and those with inadequate access to healthcare) at high risk of morbidity and mortality [11,14]. More than 5 million additional deaths per year can be attributed to abnormal temperature extremes [4]. Thus, defining vulnerable populations so they can easily be reached in an emergency climate event is urgent. Because of its latitude, altitude, continental positioning, and unique ecosystem, the Alpine region’s warming—with maximums of up to 2.5 °C warmer than usual—has been much more pronounced than in the rest of the Northern Hemisphere [15,16]. A significant proportion of Europe’s population is also exposed to air pollution levels above national and international air quality standards [5]. Ambient air pollution is the environmental factor with the greatest impact on human health; it is associated with higher mortality and morbidity [8]. Indeed, air pollution is responsible for 9 million excess deaths annually, an increase of 7% since 2015 and 66% since 2000. This is the equivalent of one in six worldwide deaths [9]. Pollution levels in Switzerland have nevertheless been able to stay below the WHO’s target in 13 out of 20 cities. The hospital admission data collected by Switzerland’s cantons provide evidence of a significant association between ambient air pollution and hospital admissions for cardiovascular and respiratory diseases [8]. Based on current knowledge, it is impossible to ensure that PM_10_ exposure does not affect hospital admissions, unlike exposure to NO_2_ and SO_2_ [8].

Climate change causes a loss of biodiversity and direct and indirect effects on physical and mental health. Its direct effects on population are increases in mental disorders, such as mood disturbances, irritability, anxiety, physical weakness, hypertension headaches, hyperalgesia, autonomic symptoms, insomnia, as well as symptoms negatively related to self-rated mental, heart failure, malnutrition, or dehydration, and sea level rise and floods, increased precipitation and mudslides, wildfires, storms, and hurricanes. Indirect effects are asthma, chronic obstructive pulmonary disease (COPD), cancers, myocardial infarction, stroke, vector-borne illnesses (dengue, malaria, Zika), and zoonoses (salmonella, brucellosis, Lyme disease). The injustice of this phenomenon is that the populations most affected are those who contribute the least to climate change. Thus, it also affects human rights and social and environmental determinants because of forced migrations, social conflicts, and increases in disparities in vulnerable populations (aboriginal population, elderly, migrants, people with chronic diseases, low-income population) [17,18,19]. To clarify the characteristics, antecedents, attributes, and consequences of climate change, it is necessary to conduct an analysis of the concepts of sustainability and exposure in environmental health.

### 1.2. Analysis of the Concept of Exposure in Environmental Health for Nursing

The field of nursing has long recognized the environment’s importance, although its definition and impact on health have varied over time. However, there was no global ecological perspective of the environment in the public-health nursing literature until the mid 1990s [15]. Today, environmental health is the promotion of human well-being through limiting exposure to hazardous agents and environmental conditions, such as water, soil, air, and food pollution [16]. Hazardous agents present in the physical environment, such as air pollution, have been identified as contributing factors to several diseases, such as ischemic heart disease, chronic obstructive pulmonary disorders, and cancer [20]. Nurses must imperatively combine their leadership roles in environmental health, health advocacy, and health interventions to raise the population’s awareness of environmental threats. Environmental health should, therefore, be incorporated into nursing education to highlight the importance of eco-literacy and ensure that the nurses of the future are aware of the environment’s impacts on health and practice in a manner that integrates both the ecological and social determinants of health. To contribute to a learning healthcare system—one that helps to grow partnerships with society, citizens, and patients—nurses’ and patients’ eco-literacy must be evaluated [21]. Social inequalities imply health inequalities [22]. Socioeconomic and demographic factors (employment, income, social support, environmental forces, media use, education), such as personal characteristics (age, race, gender, cultural background, reasoning, social skills, memory), should also be directly included in the domain of environmental health because social inequalities influence the probability to be more or less exposed to detrimental exposures, such air pollution, and contribute to people’s levels of health literacy. Low health literacy impacts health outcomes (higher risk of mortality and poorer health status among older adults, poorer ability to prove that medications are being taken appropriately, and poorer ability to interpret labels and health messages) and the use of healthcare services (increased hospitalization, less use of preventive services) [23]. It is therefore essential to determine the variables that contribute to different levels of eco-literacy [24].

Multiple disciplines—such as the industrial, epidemiological, medical, toxicological, and exposure sciences—are interested in the concept of exposure. Although it is essential to understand the links between the environment, human beings, and health, as well as environmental factors’ impacts on health, they have only very recently been addressed by the nursing discipline [25]. The exposure’s antecedents refer to hazard exposure and the places where those exposures might occur. The exposure’s attributes are classified by time and/or frequency. Finally, in terms of its consequences, exposure is described in terms of adverse health outcomes and posing a threat to human health. The potential sources of exposure are multiple: air, water, soil, food, dust, homes, pollution, or workplaces and conditions [15]. Measuring exposure must consider contact, entry, and passage. Contact—the establishment of a link between the person and one or more agents in the same environment—is measured in terms of concentration, time period, duration, frequency, and timing [15]. Exposure should be the key concept of concern when addressing environmental health issues in nursing.

### 1.3. Analysis of the Concept of Sustainability for Nursing

Using the concept of sustainability includes environmental considerations at all levels. Implementing those considerations will contribute to the development of a harm-free environment and opportunities for good health [26]. Sustainability in nursing involves five antecedents (climate change, environmental awareness, confidence in the future, responsibility, and willingness to change) and six attributes (ecology, environment, future, globalism, holism, and maintenance). The consequences of sustainability in nursing include education in the areas of ecology and the environment. The healthcare sector must participate in sustainable development. Because of its holistic perspective, the nursing sector is well positioned to influence a future of sustainable planetary health, i.e., the adoption of an eco-centric thinking [26].

### 1.4. The Environment According to Florence Nightingale

One of Florence Nightingale’s most important contributions was her environmental theory, which was developed in 1859. This focused primarily on the environment, interpreted as all the external conditions and influences that affect an organism’s life and development and prevent, suppress, or contribute to disease and death [27]. Nightingale conceptualized the environment as the place where the sick individual and/or their family members are located, considered physical, social, and psychological factors, and concluded that a healthy environment was essential to healing. Thus, the environment should be designed to promote individuals’ well-being and protect them from harm. Nightingale argued that the physical environment, including ventilation, light, noise, cleanliness, and nutrition, should be controlled to keep it healthy [27]. The nurse’s role becomes central to improving patients’ living conditions and preventing disease. Nurses’ actions must be in harmony with the five essential conditions guaranteeing a healthy environment: (i) pure air, (ii) pure water, (iii) efficient drainage, (iv) cleanliness, and (v) light [27]. For Nightingale, health went beyond the absence of disease: it was being well and being able to use all one’s powers to their full capacity. A disease is an effort of nature to restore health. She considered a human being to be a part of nature—an individual whose natural defenses were influenced by how healthy or unhealthy the environment was [27].

No other evidence is needed to demonstrate that nursing is one of the keys to dealing with the current environmental situation. Climate change’s impact has already induced significant increases in average temperatures and levels of greenhouse gases. Both its direct effects (extreme weather events, floods, drought, air pollution, extreme temperatures, worsening water quality, malnutrition, dehydration) and indirect effects (increases in allergens and other pollutants, more vector-borne diseases) [11,28,29] are directly linked to Nightingale’s five essential conditions for a healthy environment.

## 2. The Metaparadigm: Hardy’s and Fawcett’s Opposing Visions

Hardy introduced the term nursing metaparadigm in 1978 and constructed its definition using Margaret Masterman’s analysis [30] of Kuhn’s conceptualization in *The Structure of Scientific Revolutions*. Hardy defined the metaparadigm as a gestalt or a total worldview—as a way of organizing perceptions [31] and proposing an ontological conceptualization [32]. According to Hardy, at that time the nursing discipline was disordered, not because of any inability to develop empirical knowledge but because it was in a normal and necessary paradigmatic stage of its evolutionary process [31].

Fawcett stated that nursing had always been centered on fundamental concepts: the nursing metaparadigm [25]. In 1984, she appropriated the term metaparadigm somewhat differently, drawing inspiration from the writings of Yura and Torres [33] and Donaldson and Crowley [34]. Fawcett defended the idea of an epistemological structural hierarchy of knowledge. She defined the metaparadigm’s four central concepts as the most abstract components of knowledge [35]. The discipline of nursing was not in a state of confusion; on the contrary, it was at a stage of structural clarity, with, at its peak, a unifying metaparadigm that generated knowledge specific to nursing. A review of the literature outlines the consensus surrounding the nursing discipline’s four central concepts (the person receiving nursing care, the environment within which the person exists, the health–illness continuum, and nursing actions themselves) and the three specific relationships between them (person–health, person–health–environment, and person–health–nursing) that unify the nursing discipline [36].

Fawcett’s definition has guided the design of environments across the different paradigms. According to the categorization paradigm (1950–1970), phenomena are divisible into defined and isolable categories. The environment, which is physical, social, and cultural, is separate from the person. Disease is considered hostile and must be controlled. The integration paradigm (1970–1985) recognizes the multiple elements and manifestations of a phenomenon and the specific context in which it occurs. The environment is made up of various contexts (historical, social, political, etc.) within which the person or family evolves. Their interactions take the form of adaptive behaviors and are circular. Finally, the transformation paradigm (1985–today) defines the phenomenon of the environment as a global unit, interacting reciprocally and simultaneously with the world around it. The environment, composed of the entire universe, coexists with the person or family but remains distinct; it evolves at a pace in which orientation, amplitude, and speed are closely linked to past, present, and future interactions between it and humans [37].

Multiple analyses of Fawcett’s metaparadigm have revealed that it has not been helping nurses to decide what a health problem is and where to set healthcare priorities. It turns out that there is a gap, a missing link between nursing and the environment [32] even though it is an essential determinant of health. Indeed, nurses have historically focused their interventions on adapting the patient’s response to the environment rather than changing the environment [38]. Theories focusing on the relationship between the person and health describe, explain, or predict the individual’s behavior during health or illness. Theories that describe, explain, or predict individuals’ behavioral patterns—as they are influenced by environmental factors during periods of well-being and illness—are concerned with the relationship between the person, their health, and the environment. Finally, the relationship between the person, health, and nursing are addressed by theories that describe or explain nursing processes or theories that predict the effects of nursing actions [36]. What about the relationship between the environment and nursing?

There is an urgent need to establish a pragmatic role for nursing in the face of climate change. Nurses are strategically placed to respond to the impacts of climate change through their practice, research, and training [39]. Nursing education should sufficiently prepare students for the effects of climate change [20]. However, no skills are available to guide their curriculum development and nurses have low eco-literacy, i.e., awareness of environmental issues and knowledge about how to prevent them [40,41]. These parameters have contributed to the profession’s lack of commitment, feelings of demotivation, not knowing what to do, and feeling overwhelmed [42]. It also suggests why nursing research in this field has developed poorly [38]. Increased awareness about the environment by the nursing profession would contribute to planetary and population health. These would contribute to the highest levels of health, well-being, and equity through their awareness and thoughtful management of their associated political, economic, and social systems [43]. Despite the profession’s clear potential to add value on climate issues, there is a lack of investment in the domain.

Plass demonstrated the correlation between rising carbon emissions and rising temperatures in 1959 [44]; however, it was not until 2011 that the Intergovernmental Panel on Climate Change (IPCC) officially recognized this relationship [45]. Compared with the disciplines of medicine or engineering, nursing’s involvement with or recognition of climate change came late. The first publication referencing climate change in the field of nursing was in 1995, whereas it was in 1961 in medicine and 1957 in engineering (Figure 1 and Figure 2).

The environment can be conceptualized locally or in a unified manner. The local conceptualization considers the environment as static and rigid in nature [27,46,47]—a concept somewhat out of step with the implications of climate change. Indeed, this vision promotes an attitude that is disconnected from the impact that a person can have on climate change, since the only solution is to adapt to the environment rather than change it. This vision does not encourage nurses to view the environment as something their practice can affect (whether positively or negatively) or to consider the problem from a political and social viewpoint [44]. The unified conceptualization, on the other hand, considers the person and the environment as inseparable, since the environment is a fluid entity that changes in unison with the person’s perceptions [47,48,49,50]. It is also inconsistent, for several reasons, with climate change and the proactive position that the person and nursing must occupy. If the environment is only defined by a person’s perceptions, it becomes almost impossible to apprehend it, since it would be difficult to distinguish the daily weather (daily temperature fluctuations) from the gradually changing trends in temperature (long-term temperature fluctuations). Second, nurses who opt for this perspective only see the issue of climate change through their own experiences and those of their patients. This is not a problem if they are around populations who have faced extreme weather events (tornados, hurricanes, etc.) or if they have experienced them themselves. However, if this is not the case, it becomes difficult to understand and apprehend all the parameters that constitute the phenomenon [44].

## 3. Challenges for the Nursing Discipline: Development of Evidence-Based Practice toward Climate Change

No matter how the environment is conceptualized, the focus is on the individual, not on society. The accepted metaparadigm and historical conceptions of the environment have limited how nurses have related to the issue of climate change and how they have understood their professional role in adapting and mitigating the issue. A global problem such as climate change requires a broader vision. Contemporary interest in the environment has extended to the safety of the environment, climate, climate change, and planetary health [51]. Planetary health [43] considers current and future social, economic, political, and health consequences and combines local and unified conceptualizations of the environment [44]. One of the challenges for the nursing discipline of the 21st century is to detach itself from this vision of the environment that no longer corresponds to the current situation and to respond positively to the consequences of climate change for planetary health.

To encourage human adaptations and mitigations and the good positioning of the nursing discipline to climate change, it is urgent to develop and implement evidence-based practice (EBP), such as nursing interventions. EBP is defined as the integration of information from various sources to aid clinical reasoning, including the best and most recent evidence, clinical expertise, personal experiences, patient preferences, and theories underpinning nursing care [52].

The conceptual model of enhancing equity and quality: population health and health policy [53] can guide the nursing discipline in taking ownership of its new role regarding climate change challenges, while also taking into account the expertise of nurses. The model has a quadruple focus (health equity, health quality, population health, and health policy) and consists of seven multidimensional concepts (environments, population factors, population health concerns, health policy, population-centered nursologists’ activities, population quality of life, and stakeholders) [53]. In the present work, only the concepts of environment, population factors, and population-centered nursologists’ activities are reported, with a focus on health quality, population health, and health equity. In parallel, implementation science allows to understand, design, and evaluate the implementation process of evidence-based practice (EBP), guaranteeing its integration into daily care. The evaluation of people characteristics (nurses experiences, patient preferences, personal experiences), implementation strategies, and the context interacting with the implementation processes are core elements contributing to intervention designs and implementations that contribute to the quality of care and security of patients on a daily basis. The active involvement of patients and citizens, a multidisciplinary research team, or leadership is a facilitating factor to develop and implement an efficient and effective intervention and contribute to a learning healthcare system (Figure 3) [24]. These two models, as is described in Figure 3, will allow respect of the fundamental bases of EBP, which are the integration of patients and their preferences, the respect of the clinical expertise, or even personal experiences regarding climate change.

### 3.1. Environments or Upstream Factors

Environments, such as neighborhoods, education, pollution, socioeconomic status, or limited access to essential services, all have an influence on human life. The environment is conceptualized using three sub-dimensions: the physical environment (physical surroundings, including safety and climate as aspects), the socioeconomic environment (the population’s social and financial circumstances), and the cultural environment (the population’s values, beliefs, social organizations, and health practices) [53].

### 3.2. Population Factors

These are considered as social determinants of health (SDOH) and are greatly responsible for health inequities. Health outcomes are more impacted by the SDOH (80%) than by clinical care (10–20%) [54]. The SDOH are the conditions in which people are born, grow, develop, live, learn, work, and age. They are categorized into five domains: (i) neighborhood and environment (where people live, work, and play), (ii) health and quality of access to healthcare (use of health services to achieve the best health outcomes), (iii) social and community context (people with whom one communicates and connects), (iv) access to education, and (v) economic stability (financial resources). Each of these domains can affect wellness, illness, and disease conditions [53].

### 3.3. Nursologists’ Population-Centered Activities

These are actions performed by nursologists and directed at populations. They include assessment (examination of the population’s health conditions), screening (early diagnosis of illness and disease), planning, and intervention [55]. Nursologists’ activities are separated into health promotion, health restoration, and maintenance of wellness, as well as the prevention of illness and disease. Their purpose is to contribute to the population’s quality of life, which includes physical, psychological, social, economic, and environmental well-being. Population health encompasses wellness (a population’s well-being), illness (subjective and personal), and disease (objective, tangible, or measurable). The population is understood as a group of people residing in a local, state, national, or international region or an aggregate group of people with common characteristics [56].

One of the conceptual model’s propositions is that there is a relationship between nursologists’ population-centered activities and population quality of life. Thus, nurses at all levels and in all practice settings need to develop knowledge, expertise, and partnerships that will enable them to contribute to raising health to the highest level for all populations [53]. Finally, this conceptual model allows for further exploration of emancipatory knowledge [57] by adopting a political economy perspective to focus on the health system’s historical, social, economic, and political factors.

More research, guided by the conceptual model of enhancing equity and quality and the implementation science, is urgently needed to evaluate nurses’ eco-literacy, as well as the conduction of retrospective, prospective, and interventional clinical research projects to highlight the added value of the nursing discipline in responding to climate change adaptation and mitigation. This development of knowledge makes it possible to identify the nurses’ expertise and their personal experience with climate change. The same process must be carried out with the population and the patients to proactively integrate them and their preferences in the development of EBP, by considering the patient-reported experience measures (PREM) and patient-reported outcome measures (PROM) [58]. The development and implementation of EBP interventions will contribute to human adaptations and mitigations to climate change. The use of an efficient and effective implementation process will guarantee the sustainability of EBP and will improve the quality of care.

## 4. Conclusions

Nursing, health, and the environment explore the effects that environmental hazards have on planetary health. It is urgent for nurses to adapt their practice to this reality to become agents of change. Nursing as a profession faces multiple challenges, such as redefining its initial professional mandate with person-centered health promotion at its core, adopting a broader vision of the environment, taking on their leadership and responsibility, or developing nursing interventions to encourage human adaptation and mitigation regarding climate change. Another challenge for the discipline is that they must adopt a new position by evaluating the needs of the population’s health and developing, implementing, and sustaining innovative interventions, such as health promotion and disease prevention interventions to promote eco-literacy. These challenges and interventions aim to contribute to climate change adaptation and mitigation, to the evolution of nursing practice in relation to climate change, and to the transformation of the healthcare system so that it can best respond to the needs emerging from the consequences of climate change [59].

To ensure the control of the process and guarantee adequate implementation strategies, the conceptual model of enhancing equity and quality: population health and health policy combined with implementation science can be used. Indeed, effective interventions need to be conceptualized and low-value care interventions need to be de-implemented by focusing on context, people, and implementation strategies. Components should contribute to the development of EBP and to the population’s quality of life, the quality of care, patient safety, a learning healthcare system, the goal of Swiss clinical research having an impact on patient care, active patient and citizen involvement, and efficient response to the population’s needs (Figure 3) [60], a process that would address some of the challenges brought about by climate change. After such interventions are developed and their feasibility and acceptability are evaluated, an experimental study to measure their effects should be conducted.

At a more macro level, health policies need to modify the organization of healthcare services to promote climate adaptation and mitigation policies in the community, to maintain health security, to develop policies to adopt population behavior, and to support planetary health nursing practice. The aim is to contribute to a resilient healthcare system, i.e., to maintain its stability by anticipating disruptions, monitoring its functionality, and ensuring its functioning in the face of climatic change consequences [9,59].

## Figures and Tables

**Figure 1 ijerph-20-05682-f001:**
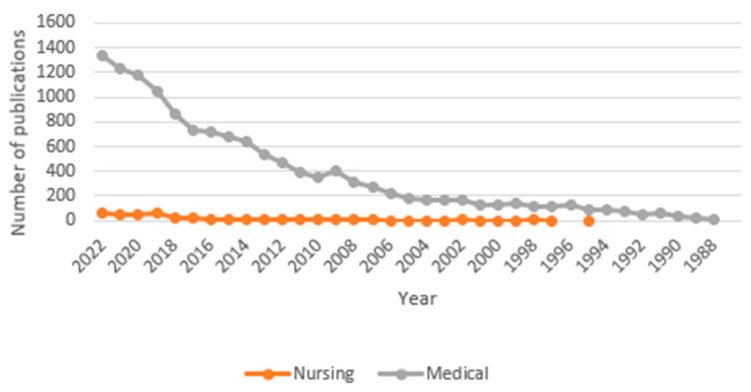
World of Science database publications since 1986 examining nursing and medicine publications on climate change. Note for nursing publications: (TS = (“Climate Chang*” OR “greenhouse effect*” OR “changing climate” OR “global warming” OR “extreme weather” OR “climate variabilit*” OR “greenhouse gas” OR “rising temperature” OR “heat wave*” OR “Environmental Pollution” OR “Air Pollution*”)) **AND** (SU = (Nursing)). Note for medical publication: (TS = (“Climate Chang*” OR “greenhouse effect*” OR “changing climate” OR “global warming” OR “extreme weather” OR “climate variabilit*” OR “greenhouse gas” OR “rising temperature” OR “heat wave*” OR “Environmental Pollution” OR “Air Pollution*”)) **AND** (SU = (“Critical Care Medicine” OR “Emergency Medicine” OR Oncology OR Physiology OR “Research & Experimental Medicine” OR “Respiratory System” OR Surgery OR “Gastroenterology & Hepatology” OR “General & Internal Medicine” OR “Geriatrics & Gerontology” OR “Infectious Diseases” OR “Medical Ethics” OR “Medical Informatics”)).

**Figure 2 ijerph-20-05682-f002:**
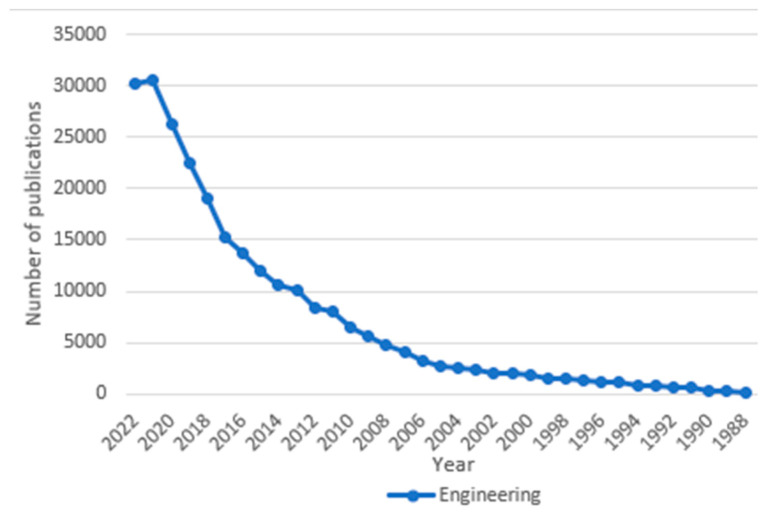
World of Science database publications since 1986 examining engineering publications on climate change. Note: (TS = (“Climate Chang*” OR “greenhouse effect*” OR “changing climate” OR “global warming” OR “extreme weather” OR “climate variabilit*” OR “greenhouse gas” OR “rising temperature” OR “heat wave*” OR “Environmental Pollution” OR “Air Pollution*”)) **AND** (SU = (“Meteorology & Atmospheric Sciences” OR “Environmental Sciences & Ecology” OR “Urban Studies”)).

**Figure 3 ijerph-20-05682-f003:**
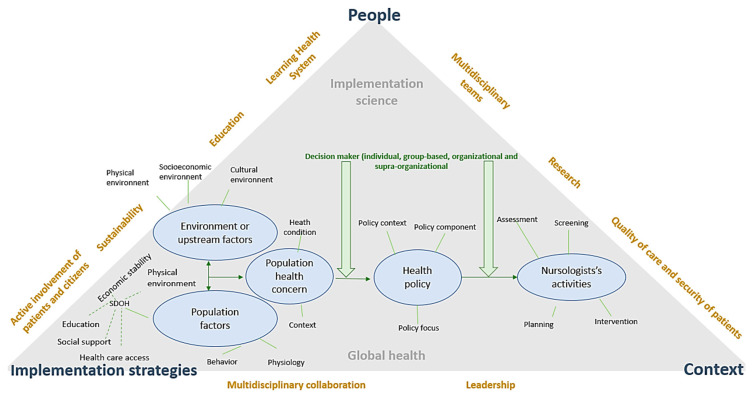
Concepts and relationships inspired by “The Conceptual Model of Nursology for Enhancing Equity and Quality: Population Health and Health Policy” [53] and implementation science [21].

## Data Availability

Not applicable.

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
