# Peer review of "Climate Change, Environmental Health, and Challenges for Nursing Discipline"

_ijerph, 2023, doi:10.3390/ijerph20095682_

Round 1

Reviewer 1 Report

Abstract missing from the paper

Line 43 to 45, is these are indicator of climate change

Line 43 to 48 need more citation to justify the points made.

How climate change is a challenge for nursing disciple, paper is unable to justify the point.

English need minor revision 

Reviewer 2 Report

The article provides a comprehensive analysis of the development of nursing knowledge and its evolution over time. The paper is based on a sound theoretical framework and the authors have used appropriate references to support their arguments. Besides, it discusses the challenges faced by the nursing profession and provides recommendations for future research and practice. Overall, the paper is a significant contribution to the field of nursing and provides valuable insights into the evolution of nursing knowledge. The following are some general comments that may help the authors to improve the quality of the article:

·       The authors could provide more specific examples of how climate change is affecting human health and how nurses can address these issues. This would help to make the article more engaging and relevant to readers.

·       While the authors have included relevant references, some of them are quite old. It would be helpful to include more recent studies and reports to ensure that the information presented is up-to-date.

·       The authors could provide more detail on the specific actions that nurses can take to address the health impacts of climate change. This would help to make the article more actionable and provide readers with practical steps they can take.

·       The article focuses primarily on the impact of climate change on health in developed countries. It would be helpful to include more information on how climate change is affecting health in developing countries and how nurses can address these issues.

·       The article ends somewhat abruptly without a clear conclusion. The authors could provide a summary of the key points and recommendations for future research and action

Reviewer 3 Report

1.          Based on the manuscript, it seems that the manuscript lacks appropriate academic analysis and theoretical or practical reflections on the issue at hand. To improve the manuscript, the author should consider conducting further research and analysis to support their arguments and provide more in-depth insights into the topic.

2.          The author should expand on the advantages and disadvantages of nursing as a discipline, as this is relevant to the topic and would provide more comprehensive insights into the field.

3.          In the introduction section, the author should provide more information about the methodological or analytical discourse they used to explore the issue. This could include a description of the research methods used, the theoretical frameworks applied, or the analytical tools employed to analyze the data.

4.          In the conclusion section, the author should summarize the key findings and insights from the manuscript, and provide theoretical reflections and practical suggestions based on the content. This could include recommendations for future research, implications for nursing practice, or suggestions for policy changes in the field.

Round 2

Reviewer 3 Report

The author mostly incorporated my suggestions and revised the manuscript. The manuscript could be published in this form for the journal.